# Coherent Entity Disambiguation via Modeling Topic and Categorical Dependency

**Zilin Xiao**♠ **Linjun Shou**♣ **Xingyao Zhang**♣
**Jie Wu**♣ **Ming Gong**♣ **Daxin Jiang**♣
Rice University♠ Microsoft STCA♣
zilin@rice.edu
{lisho, xingyaozhang, jiewu1, migon, djiang}@microsoft.com

## Abstract

Previous entity disambiguation (ED) methods adopt a discriminative paradigm, where prediction is made based on matching scores between mention context and candidate entities using length-limited encoders. However, these methods often struggle to capture explicit discourse-level dependencies, resulting in incoherent predictions at the abstract level (*e.g.* topic or category). We propose COHERENTED, an ED system equipped with novel designs aimed at enhancing the coherence of entity predictions. Our method first introduces an unsupervised variational autoencoder (VAE) to extract latent topic vectors of context sentences. This approach not only allows the encoder to handle longer documents more effectively, conserves valuable input space, but also keeps a topic-level coherence. Additionally, we incorporate an external category memory, enabling the system to retrieve relevant categories for undecided mentions. By employing step-by-step entity decisions, this design facilitates the modeling of entity-entity interactions, thereby maintaining maximum coherence at the category level. We achieve new state-of-the-art results on popular ED benchmarks, with an average improvement of 1.3 F1 points. Our model demonstrates particularly outstanding performance on challenging long-text scenarios.

## 1 Introduction

Entity disambiguation (ED) is a typical knowledge-intensive task of resolving mentions in a document to their corresponding entities in a knowledge base (KB), *e.g.* Wikipedia. This task is of great importance due to its active presence in downstream tasks such as information extraction (Hoffart et al., 2011), question answering (Yih et al., 2015) and web search query (Blanco et al., 2015).

To perform efficient entity disambiguation (ED), one common approach is to encode mentions and candidate entities into different embedding spaces .

Then a simple vector dot product is used to capture the alignment between mentions and candidate entities. While this method enables quick maximum inner product search (MIPS) over all candidates and efficiently determines the linked answer, it suffers from late and simplistic interaction between mentions and entities (Barba et al., 2022; Cao et al., 2021). Recently, researchers have proposed alternative paradigms for solving the ED problem, such as formulating it as a span extraction task (Barba et al., 2022). In this approach, a Longformer (Beltagy et al., 2020) is fine-tuned to predict the entity answer span within a long sequence consisting of the document and candidate entity identifiers. Another paradigm (Cao et al., 2021; De Cao et al., 2022) reduces the ED task to an auto-regressive style in which generation models are trained to produce entity identifiers token-by-token.

Although these approaches offer some mitigation for the late-and-simple interaction problem, they still exhibit certain vulnerabilities. For instance, Transformer-based encoders impose inherent limitations on input length, preventing the capture of long-range dependency for specific mentions. Also, these methods do not explicitly consider coherence constraints, while coherence is considered as important as context in early ED works (Hoffart et al., 2011; Chisholm and Hachey, 2015). We first propose to condition the model on compressed topic tokens, enabling the system to sustain **topic coherence** at the document level.

In addition, the relationship among entities holds significant importance in the ED task. For example, mentions in the document exhibit a high correlation at the category level, where we name it **category coherence**. However, previous bi-encoder and cross-encoder solutions have overlooked these entity dependencies and focused solely on learning contextualized representations. Among other works, extractive paradigm (Barba et al., 2022) neglects entity-entity relation as well; generative

EL (Cao et al., 2021; De Cao et al., 2022) do possess some dependencies when linking an unknown mention. However, these dependencies arise from the auto-regressive decoding process and require heavy inference compute.

To address the above coherence problem, we propose two orthogonal solutions that target topic coherence and entity coherence, respectively. Following previous works that decode masked tokens to link unknown mentions (Yamada et al., 2020, 2022), we present the overview of our coherent entity disambiguation work in Figure 1, where document words and unresolved entities are treated as input tokens of Transformer (Vaswani et al., 2017). First, we bring an unsupervised variational auto-encoder (VAE) (Kingma and Welling, 2014) to extract topic embeddings of surrounding sentences, which are later utilized to guide entity prediction. By docking these two representative language learners, BERT (Devlin et al., 2019) and GPT-2 (Radford et al., 2019), the variational encoder can produce topic tokens of sentences without training on labeled datasets. This approach promotes a higher level of coherence in model predictions from an abstract level (Li et al., 2020) (*e.g.* tense, topic, sentiment).

Moreover, in most KBs, categories serve as valuable sources of knowledge, grouping entities based on similar subjects. To enhance entity-entity coherence from a categorical perspective, we design a novel category memory bank for intermediate entity representations to query dynamically. As opposed to retrieving from a frozen memory layer, we introduce direct supervision from ground-truth category labels during pre-training. This enables the memory to be learned effectively even from random initialization.

Named COHERENTED, experimental results show that our proposed methods surpass previous state-of-the-art peers on six popular ED datasets by 1.3 F1 points on average. Notably, on the challenging CWEB dataset, which has an average document length of 1,700 words, our approach elevates the score from the previous neural-based SOTA of 78.9 to 81.1. Through model ablations, we verify the effectiveness of the two orthogonal solutions through both quantitative performance evaluation and visualization analysis. These ablations further affirm the superiority of our methods in generating coherent disambiguation predictions.

## 2 Related Works

**Entity disambiguation (ED)** is a task of determining entities for unknown mention spans within the document. Early ED works (Hoffmann et al., 2011; Daiber et al., 2013) commonly rely on matching scores between mention contexts and entities, disregarding the knowledge-intensive nature of ED. Many studies aim to infuse external knowledge into ED. Bunescu and Paşca (2006) begin to utilize hyperlinks in Wikipedia to supervise disambiguation. Yamada et al. (2020, 2022) propose massive pre-training on paired text and entity tokens to implicitly inject knowledge for disambiguation usage. Li et al. (2022) first leverage knowledge graphs to enhance ED performance.

Another intriguing feature for entities to differentiate from each other is their type, as entities with similar surface forms (text identifiers) often possess different types. Onoe and Durrett (2020) disambiguate similar entities solely through a refined entity type system derived from Wikipedia, without using any external information. Ayoola et al. (2022) augment the robustness of such a type system even further. Furthermore, many researchers have explored new paradigms for ED. Cao et al. (2021) propose using a prefix-constrained dictionary on casual language models to correctly generate entity strings. Barba et al. (2022) recast disambiguation as a task of machine reading comprehension (MRC), where the model selects a predicted entity based on the context fused with candidate identifiers and the document.

The architecture of our system seamlessly blends the advantages of type systems and knowledge pre-training: incorporating type systems as dynamically updating neural blocks within the model, our design enables simultaneous learning through a multi-task learning schema - facilitating topic variation learning, masked disambiguation learning, and knowledge pre-training concurrently.

**Prompt Compression** is a commonly used technique in language models to economize input space, closely related to topics such as prompt compression (Wingate et al., 2022) and context distillation (Snell et al., 2022). Their mutual goal is to dynamically generate soft prompt tokens that replace original tokens without hurting downstream application performance. Our topic token design mirrors context compression to some extent but differs in the compression ratio and purpose. Our design is more compact; each context sentence gets

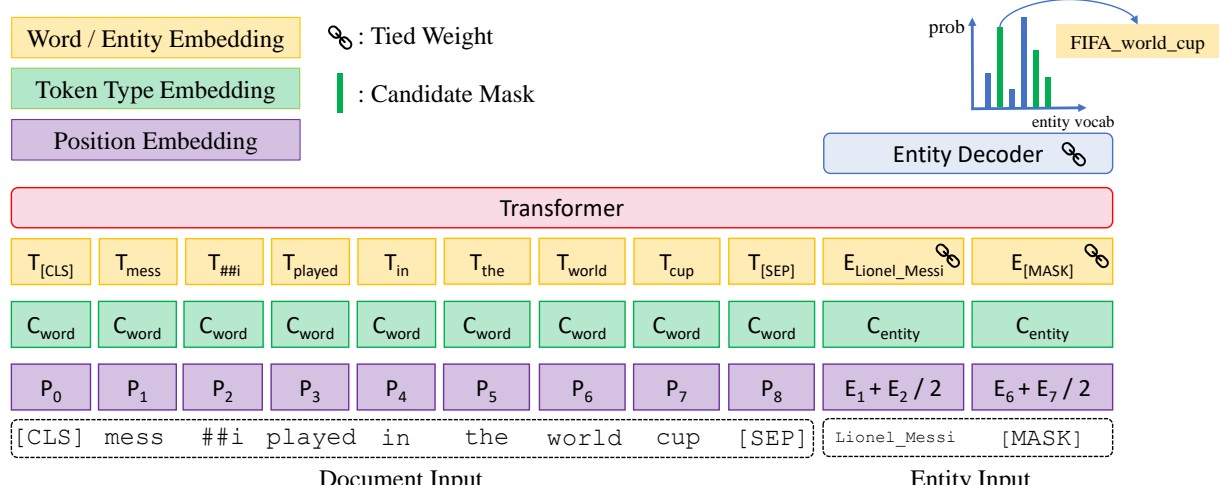

Figure 1: Overview of our baseline model. Legends are presented in the upper left corner. Position embedding for an entity is averaged on corresponding document positions, for example, $E_1 + E_2 / 2$ indicates the entity "Lionel_Messi" is predicted based on document tokens `mess` and `##i` at position 1 and 2.

converted into a single topic token using a variational encoder, and in addition to saving input space, it also retains high-level semantics to guide a more coherent ED.

## 3 Methodology

### 3.1 Entity Disambiguation Definition

Let $X$ be a document with $N$ mentions $\{m_1, m_2, \ldots, m_N\}$, where each of mentions $m_i$ is associated with a set of entity candidates $C_i = \{e_{i1}, e_{i2}, \ldots, e_{i|C_i|}\}$. Given a KB with a set of triplets $G = \{(h, r, t)|h, t \in \mathcal{E}, r \in \mathcal{R}\}$, where $h$, $r$ and $t$ denote the head entity, relation and tail entity respectively, the goal of entity disambiguation is to link mentions $m_i$ to one of the corresponding mention candidates $C_i \subseteq \mathcal{E}$.

### 3.2 Overview

We present the overview of COHERENTED in Figure 1. Following Yamada et al. (2020), both words in the document and entities are considered input tokens for the BERT model. The final input representation sums over the following embeddings:

**Representation embedding** denotes the topic latent, word embedding or entity embedding for topic inputs, document inputs or entity inputs accordingly. We set up two separate embedding layers for word and entity input respectively. $\mathbf{X} \in \mathbb{R}^{V_w \times H}$ denotes the word embedding matrices and $\mathbf{Y} \in \mathbb{R}^{V_e \times H}$ denotes the entity embedding matrices, where $V_w$ and $V_e$ represents the size of word vocabulary and entity vocabulary.

**Type embedding** is for discrimination usage. There are three types of tokens available, each of which corresponds to a dedicated group of parameters, $\mathbf{C}_{word}, \mathbf{C}_{entity}, \mathbf{C}_{topic}$.

**Position embedding** marks the position of input words and entities, avoiding the permutation-invariant property of the self-attention mechanism. The entity position embedding also indicates which word tokens the entity corresponds to. This is achieved by applying absolute position embedding to both words and entities. If a mention consists of multiple tokens, the entity position embedding is averaged over all corresponding positions.

### 3.3 Topic Variational Autoencoder

To preserve maximum topic coherence and optimize input space utilization, we introduce an external component that facilitates topic-level guidance in ED prediction. Among various latent variable models, variational autoencoders (VAEs) have demonstrated success in modeling high-level syntactic latent factors such as style and topic. Our setup is based on the motivation that, after being trained on a massive corpus using the variational objective, the encoder gains the capacity to encode sentences into topic latent vectors. The entire topic VAE is composed of two parts, BERT and GPT-2, both of which are powerful Transformer-based language encoder and decoder.

#### 3.3.1 Encoder

Given a BERT encoder $LM_\phi$ and input sentence token sequence $x$, we collect the aggregated em-

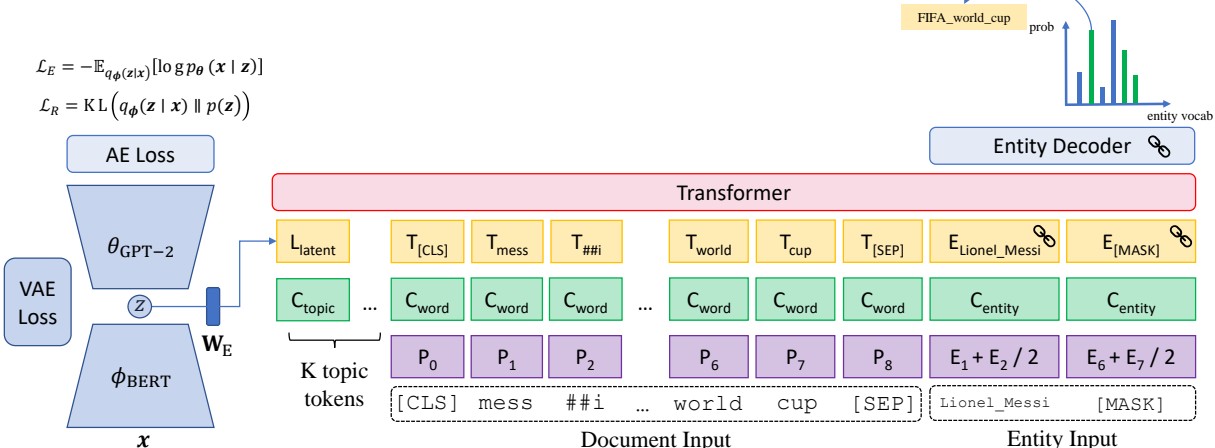

Figure 2: Topic variational autoencoder injection illustration. $K$ topic sentences are converted into topic tokens appending on the start of model's input.

bedding $X$ from the last layer hidden states corresponding to the [CLS] token: $X = \text{BERT}_{\text{[CLS]}}(x)$. With this, we can construct a multivariate Gaussian distribution from which the decoder will draw samples. The following formula describes the variational distribution for the approximation of the posterior:

$$q_\phi\left(z_n \mid \boldsymbol{x}_{\leq n}, \boldsymbol{z}_{<n}\right) = \mathcal{N}\left(z_n \mid f_{\mu_\phi}\left(x_n\right), f_{\sigma_\phi}\left(x_n\right)\right), \quad (1)$$

where $f_{\mu_\phi}$ and $f_{\sigma_\phi}$ denote separate linear layers for mean and variance representations, $\mathcal{N}$ denotes Gaussian distribution and $z$ denotes intermediate information bottleneck.

### 3.3.2 Decoder

Given a GPT-2 decoder $\text{LM}_\theta$, we first review how to generate a text sequence of length $T$ using such neural models. To generate word tokens of length $L$, $\mathbf{x} = [x_1, x_2, \ldots, x_T]$, a language decoder utilizes all its parameters $\theta$ to predict the next token conditioned on all previous tokens generated $x_{<t}$, formulated as follows:

$$p(\boldsymbol{x}) = \prod_{t=1}^{T} p_{\boldsymbol{\theta}}\left(x_t \mid x_{<t}\right). \quad (2)$$

When training a language generator alone, the decoder is usually learned using the maximum likelihood estimate (MLE) objective. However, in our VAE setting, the decoder conditions on the vector $z$ dynamically drew from a Gaussian distribution, instead of purely from previously generated

tokens. Specifically, our decoder generates autoregressively via:

$$p_{\boldsymbol{\theta}}(\boldsymbol{x} \mid \boldsymbol{z}) = \prod_{t=1}^{T} p_{\boldsymbol{\theta}}\left(x_t \mid x_{<t}, \boldsymbol{z}\right). \quad (3)$$

As stated above in Part 3.3.1, the intractable posterior for $z_n$ is approximated by $q_\phi\left(z_n \mid \boldsymbol{x}_{\leq n}, \boldsymbol{z}_{<n}\right)$ in Equation 1. Now we see the difference in the VAE decoder: the generation relies on high-level semantics and has the ability to produce a compact representation.

### 3.3.3 ELBO Training

Both the encoder and decoder need to be trained to optimize their parameters. Supported by the above approximation, the entire training objective can be interpreted as evidence lower bound objective (ELBO):

$$\log p_{\boldsymbol{\theta}}(\boldsymbol{x}) \geq \mathcal{L}_{\text{ELBO}} =$$
$$\mathbb{E}_{q_\phi(\boldsymbol{z}|\boldsymbol{x})}\left[\log p_{\boldsymbol{\theta}}(\boldsymbol{x} \mid \boldsymbol{z})\right] - \text{KL}\left(q_\phi(\boldsymbol{z} \mid \boldsymbol{x}) \| p(\boldsymbol{z})\right).$$

Detailed derivations are emitted for clear idea depiction. In practice, we apply the reparametrization trick (Kingma and Welling, 2014) to allow back-propagation through all deterministic nodes and for efficient learning.

Intuitively, we consider the first term as an autoencoder objective, since it requires the model to do reconstruction based on the intermediate latent. The second term defines the KL divergence between the real distribution $q_\phi(\boldsymbol{z} \mid \boldsymbol{x})$ and $p(\boldsymbol{z})$. To better implement these objectives, we refer to

the regularized version of ELBO (Li et al., 2020), where ELBO is considered as the linear combination of reconstruction error and KL regularizer:

$$\mathcal{L}_{\text{variational}} = \mathcal{L}_E + \beta\mathcal{L}_R, \text{ with}$$
$$\mathcal{L}_E = -\mathbb{E}_{q_\phi(\boldsymbol{z}|\boldsymbol{x})}\left[\log p_{\boldsymbol{\theta}}(\boldsymbol{x} \mid \boldsymbol{z})\right] \quad (4)$$
$$\mathcal{L}_R = \text{KL}\left(q_\phi(\boldsymbol{z} \mid \boldsymbol{x}) \| p(\boldsymbol{z})\right).$$

Finally, we treat the loss term $\mathcal{L}_{\text{variational}}$ as one of our minimization objectives in the following multi-task learning.

### 3.4 Category Memory

We now formally define the category memory layer inserted into the intermediate Transformer layers. Let $\mathcal{E} = \{e_1, e_2, \ldots, e_n\}$ be the set of all possible candidate entities, and correspondingly, each entity $e_i$ has a predefined set of categories $C_{e_i}$. The union set of category sets of all candidate entities will be $C$. Based on this motivation, we construct a category vocabulary and a category embedding table $\mathbf{C} \in \mathbb{R}^{|C| \times d_{\text{category}}}$. The vocabulary establishes a mapping from textual category labels to indices. As sometimes category labels can be too fine-grained, we refer readers to Appendix A for the detailed design of category system.

The embedding table $\mathbf{C}$ stores category representations that can be updated during massive pre-training. To be specific, we first formulate our model's forward (Figure 3) as follows:

$$\mathbf{T}^1, \mathbf{W}^1, \mathbf{E}^1 = \text{Transformer}_M\left(\mathbf{T}^0, \mathbf{W}^0, \mathbf{E}^0\right),$$
$$\mathbf{H} = \text{CategoryMemory}\left(\mathbf{E}^1\right),$$
$$\mathbf{E}^{1\prime} = \text{LayerNorm}\left(\mathbf{H} + \mathbf{E}^1\right),$$
$$\mathbf{T}^2, \mathbf{W}^2, \mathbf{E}^2 = \text{Transformer}_N\left(\mathbf{T}^1, \mathbf{W}^1, \mathbf{E}^{1\prime}\right),$$

where symbols $\mathbf{T}$, $\mathbf{W}$, $\mathbf{E}$ stand for hidden states of topics, words and entities respectively.

Each unresolved entity in the document is assigned with [MASK] token in the input sequence. During the forward pass, all intermediate entity representations corresponding to [MASK] token $e^i_{\text{masked}}$ are projected from $\mathbb{R}^{d_{\text{entity}}}$ to $\mathbb{R}^{d_{\text{category}}}$ using a linear layer without bias terms:

$$\mathbf{E}^i_{masked} = \mathbf{W_A} \cdot e^i_{\text{masked}} \quad (5)$$

Subsequently, the adapted intermediate entity representations will query all entries in the category embedding table. The aggregated weighted hidden states are computed as follows:

$$\mathbf{H}^{\text{category}}_s = \mathbf{W_B}\left(\sum_{j=1}^{|C|} \alpha_{ij} \cdot \mathbf{C}^j\right), \quad (6)$$

where $\alpha_{ij} = \text{sigmoid}(\mathbf{C}^j \cdot (\mathbf{E}^i_{\text{masked}})^T)$ denotes matching score between $i$-th masked entity and $j$-th category. $\mathbf{W_B}$ is a linear projection layer for dimension matching. During training, we apply direct supervision from gold category guidance via binary cross-entropy loss:

$$\mathcal{L}_{\text{category}} = -\frac{1}{|C|}\sum_{j=1}^{|C|} \alpha_{ij} \cdot \mathbb{I}_{\text{oracle}}, \quad (7)$$

where $\mathbb{I}_{\text{oracle}}$ denotes the indicator function of the oracle category labels for the $i$-th masked entity.

### 3.5 Multi-task Pre-training

This part discusses the pre-training stage for Co-HERENTED. First, we define the disambiguation loss, which is analogous to the well-known masked language modeling objective. In each training step, 30% of the entity tokens are replaced with a special [MASK] token. We employ a linear decoder at the end of our model to reconstruct the masked tokens, as shown in Equation 8.

$$\hat{\mathbf{E}} = \text{softmax}(\mathbf{W_D} \cdot \mathbf{E}^2_{masked} + \mathbf{b_D}). \quad (8)$$

Equation 9 represents the cross entropy loss over the entity vocabulary, where $\mathbb{I}_{e_k}$ denotes the indicator function of the $k$-th masked entity's ground-truth.

$$\mathcal{L}_{\text{disambiguation}} = -\frac{1}{N_{\text{masked}}}\sum_{k=1}^{N_{\text{masked}}} \hat{\mathbf{E}} \cdot \mathbb{I}_{e_k}. \quad (9)$$

By incorporating all these losses, we derive our final multi-task learning objective being:

$$\mathcal{L} = \mathcal{L}_{\text{disambiguation}} + \alpha\mathcal{L}_{\text{variational}} + \gamma\mathcal{L}_{\text{category}}, \quad (10)$$

where coefficients $\alpha$ and $\gamma$ control relative importance of two auxiliary tasks.

### 3.6 COHERENTED Inference

Given a document $X$ with $N$ mentions, $M = \{m_1, m_2, \ldots, m_N\}$, we now describe the coherent ED inference process.

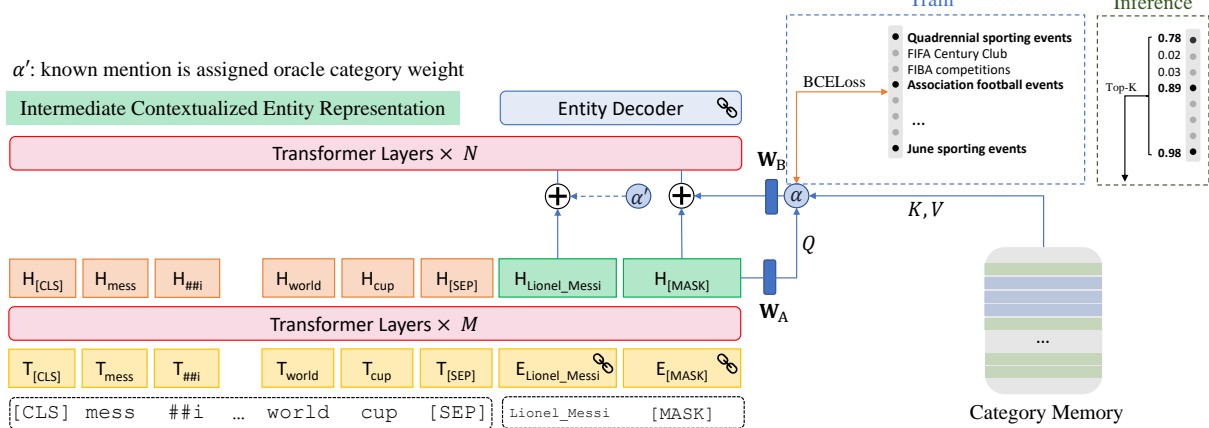

Figure 3: Category memory illustration. The difference between training and inference is depicted using two dotted colored box.

Considering a Transformer with an input length limit being $L$ tokens, we reserve $k$ tokens for topic latent, $n_e$ for shallow entity representation input, leaving the word input window being $L - k - n_e$ tokens. Note that $n_e$ indicates the number of mentions in a certain sentence and varies among different training batches, so we set $n_e$ to the maximum number of mentions within the batch. We refer interested readers to Appendix C for how we sample topic sentences and prepare input tokens.

With all input tokens ready, we predict entities for $N$ steps. Unlike a language generator which decodes the next token, at each step $i$, the model decodes all [MASK] tokens into entity predictions by selecting maximum indices in the logits. The entity prediction at step $i$ is decided using the highest-confidence strategy, *i.e.*, the entity with the highest log probability is resolved, while others have to wait until the next step.

It is worth noting that utilizing candidate entity information can significantly reduce noisy predictions, as indicated by the green bars in Figure 1. During inference in the category memory layer, only top-$k$ category entries are selected for weighted aggregation.

Once an entity prediction is determined, the category memory ceases to be queried at that position and instead receives a real category indicator to aggregate entries from the category memory. We refer to this as **oracle category guidance** because it allows for potential category-level guidance in disambiguating remaining mentions.

## 4 Experiments

### 4.1 Datasets and Settings

For a fair comparison with previous works, we adopt the exact same settings used by Cao et al. (2021). Specifically, we borrow pre-generated candidate entity sets from (Le and Titov, 2018). Only entities with the top 30 $\hat{p}(e \mid m)$ score are considered candidates, and failure to include the oracle answer in the candidate set leads to a false negative prediction. For evaluation metrics, we report *InKB* micro F1 scores on test splits of AIDA-CoNLL dataset (Hoffart et al., 2011) (AIDA), cleaned version of MSNBC, AQUAINT, ACE2004, WNED-CWEB (CWEB) and WNED-WIKI (WIKI) (Guo and Barbosa, 2018). For training data, we use the 2022-09-01 Wikipedia dump without any weak or pseudo labels utilized in Orr et al. (2021); Broscheit (2019). We do not run hyperparameter search due to limited training resources. Note that a few previous works use a mixture of AIDA and Wikipedia as training split, which we will indicate in the table caption as an unfair setting. Refer to Appendix B for dataset details, and Appendix E for implementation, training and hyperparameter choice.

Two variants are proposed for a fair comparison with baseline models in terms of the number of Transformer layers. COHERENTED_base contains 3, 3, and 6 layers for the VAE encoder, decoder and base model respectively, while in COHERENTED_large these numbers become 6, 6 and 12. Both variants are with a comparable number of parameters with Yamada et al. (2022), namely 210M for the base variant and 440M for the large one.

| Method | AIDA | MSNBC | AQUAINT | ACE2004 | CWEB | WIKI | AVG-6 | AVG-5 |
|---|---|---|---|---|---|---|---|---|
| Yosef et al. (2011) | 78.0 | 79.0 | 56.0 | 80.0 | 58.6 | 63.0 | 69.1 | 67.3 |
| van Hulst et al. (2020) | 89.4 | 90.7 | 84.1 | 85.3 | 71.9 | 73.1 | 82.4 | 81.0 |
| Cao et al. (2021) | **93.3**$^\dagger$ | 94.3 | 89.9 | 90.1 | 77.3 | 87.4 | 88.7$^\dagger$ | 87.8 |
| Orr et al. (2021)* | 80.9 | 80.5 | 74.2 | 83.6 | 70.2 | 76.2 | 77.6 | 76.9 |
| Yang et al. (2018) | 93.0$^\dagger$ | 92.6 | 89.9 | 88.5 | **81.8** | 79.2 | 87.5$^\dagger$ | 86.4 |
| Barba et al. (2022) | 92.6$^\dagger$ | 94.7 | 91.6 | 91.8 | 77.7 | 88.8 | 89.5$^\dagger$ | 88.9 |
| Ayoola et al. (2022) | 87.5 | 94.4 | 91.8 | 91.6 | 77.8 | 88.7 | 88.6 | 88.8 |
| Yamada et al. (2022) | - | **96.3** | 93.5 | 91.9 | 78.9 | 89.1 | - | 89.9 |
| Our CoherentED$_{base}$ | 88.2 | 94.9 | 93.7 | 92.3 | 77.2 | 87.8 | 89.0 | 89.2 |
| Our CoherentED$_{large}$ | 89.4 | **96.3** | **94.6** | **93.4** | 81.1 | **90.6** | **90.9** | **91.2** |
| Model Ablations on CoherentED$_{large}$ | | | | | | | | |
| - w/o Topic Tokens | 88.4 | 94.6 | 93.4 | 92.3 | 77.9 | 88.9 | 89.3 | 89.4 |
| - w/o Category Memory | 89.1 | 95.6 | 93.8 | 92.3 | 79.8 | 90.1 | 90.1 | 90.3 |
| - w/o Category Oracle Guidance | 89.9 | 95.2 | 92.1 | **93.4** | 80.2 | **90.8** | 90.3 | 90.3 |

Table 1: ED InKB micro F1 scores on test datasets. The best value is in **bold** and the second best is in underline. * means results come from reproduced results of official open-source code. $^\dagger$ indicates non-comparable metrics due to an unfair experimental setting. - indicates not reported in the original paper. For direct comparison with Yamada et al. (2022), AVG-5 reports average micro F1 scores on all test datasets except AIDA.

## 4.2 Main Results

We report peer comparisons in Table 1. Note that we consider the Wikipedia-only training setting, meaning no further fine-tuning or mixture training on AIDA is allowed and all evaluations are out-of-domain (OOD) tests.

In general, we achieve new state-of-the-art results on all test datasets except CWEB and AIDA datasets, surpassing the previous best by 1.3 F1 points and eliminating 9% errors. On the CWEB dataset, our work still shows superiority over other neural-based methods, as the lengthy samples are too unfriendly to be understood globally by native neural encoders. On the ACE2004 dataset, since the number of mentions is relatively small, many reported numbers are identical. On other datasets, the relative improvements are consistent for COHERENTED$_{large}$ even though only additional cheap category labels are provided during pre-training. Such improvements also confirm the outstanding OOD ability of our methods since no fine-tuning is conducted on the downstream test datasets.

## 4.3 Ablation Study

In the lower part of Table 1, we report ablation experiments of proposed methods. All ablations are conducted on COHERENTED$_{large}$.

Compared with the model without topic token injections, COHERENTED$_{large}$ improves greatly from 89.3 to 90.9 in average micro F1 score with a particular gain on the lengthy CWEB dataset from 77.9 to 81.1. Such performance gain on CWEB only falls short of Yang et al. (2018) which requires extensive feature engineering targeted at document-level representations. All other test sets also benefit from the powerful abstract modeling ability of topic VAE.

Compared with the model without a category memory layer, the CoherentED$_{large}$ improves from 90.1 to 90.9, not as significant as the improvement of topic injections but still notable enough. Note that category memory does not bring consistent performance gain on all test datasets, possibly because not all test samples are sensitive to category-level coherence. Furthermore, we attempt to disable the category oracle guidance strategy during evaluation, meaning for predicted entities in step-by-step ED, we still query the external category memory and aggregate the retrieved entries. And ablation shows that the oracle guidance does have a positive impact on the overall performance metrics.

## 4.4 Case Analysis and Visualization

Besides ablations on evaluation metrics, we conduct a deeper analysis of proposed methods by visualizing data samples. Specifically, we per-

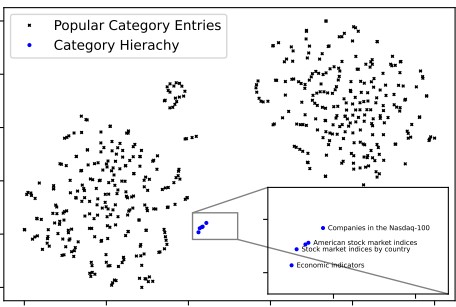

Figure 4: T-SNE visualization of category embeddings after pre-training. Zoomed part denotes the specific category labels and corresponding sample points. Category "Stock market indices" is emitted for clear depiction.

form t-SNE on category memory entries after joint training and topic vectors of sentences in MSNBC dataset.

To validate the effectiveness of the learned category memory layer, we expect the embeddings of category entries stored in the memory to exhibit a certain degree of similarity with the category hierarchy in Wikipedia, *i.e.*, similar category entries are close to each other. In Figure 4, we present t-SNE visualization of the top 500 popular category entries and additional colored category group, where black cross data points represent popular category entries and blue circular points represent examples of a structured hierarchy from the Wikipedia category system[1].

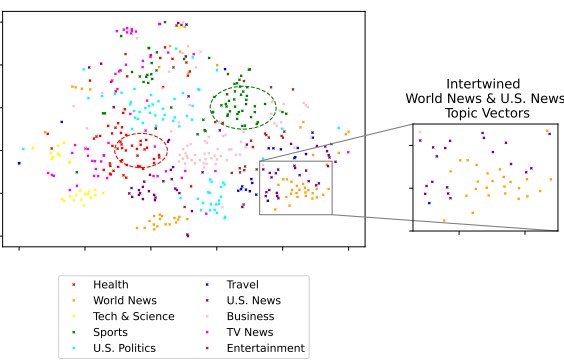

Figure 5: T-SNE visualization of sentence vectors extracted by the topic probe strategy. Some polarized topic groups such as "Health" and "Sports" are denoted using colored dashed circles. Zoomed part reveals the intertwined nature between similar topics (*e.g.* "World News" and "U.S. News"). Best viewed in color.

To evaluate the topic representation ability of our jointly trained topic VAE, we design an elegant probe strategy to investigate the topic modeling ability of COHERENTED. MSNBC covers 20 documents on 10 topics[2]. By feeding each sentence in the MSNBC test set along with predicted entities tokens, we extract the [CLS] representation of CoherentED$_{large}$ and run t-SNE on these joint representations.

In Figure 5, topic latent vectors of sentences in these documents are plotted into 511 data points, whose colors denote their oracle topic labels. We see that the majority of sentences under the same topic cluster into polarized groups, despite a few outliers possibly because they are for general purposes such as greeting and describing facts. Consistent with our expectations, similar topics are intertwined as they share high-level semantics to some extent.

## 5 Conclusion

We propose a novel entity disambiguation method COHERENTED, which injects latent topic vectors and utilizes cheap category knowledge sources to produce coherent disambiguation predictions. Specifically, we introduce an unsupervised topic variational auto-encoder and an external category memory bank to mitigate inconsistent entity predictions. Experimental results demonstrate the effectiveness of our proposed methods in terms of accuracy and coherence. Analysis of randomly picked cases and vector visualizations further confirm such effectiveness.

## Limitations

Still, our COHERENTED remains with two limitations: scalability and performance. Future works are expected to alleviate these limitations. First, COHERENTED can hardly handle emerging entities as this requires extending both the entity embedding layer and category memory layer. The evaluation metrics will degrade if no further training is conducted after such expansion. Second, despite the count of parameters and FLOPs of CO-HERENTED being quite comparable with baseline models, the advantage of coherent prediction only reveals itself in the scenario of step-by-step reasoning, *i.e.*, mentions are resolved one by one. This

---

[1]Economic indicators → Stock market indices → American stock market indices → Companies in the Nasdaq-100

[2]Business, U.S. Politics, Entertainment, Health, Sports, Tech & Science, Travel, TV News, U.S. News, and World News.

means multiple forward passes are needed for each document to achieve the most accurate results.

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

## A  Category System Design

Sometimes category labels can be too fine-grained, *e.g.* Apple Inc. is with one category label of Computer companies established in 1976, which is less informative compared with two separate labels Computer companies and in 1976. To mitigate this issue, we extend category construction methods from Onoe and Durrett (2020), where prepositions [3] are considered as stop words to disassemble original category labels. We further ignore differences among prepositions, *i.e.*, of the United States and in the United States are considered to be indistinguishable and unified into [PERP] the United States. This helps simplify the category labels and ensures that the model focuses on the relevant semantic information rather than specific prepositions.

## B  Evaluation Dataset Details

The brief dataset descriptions are as follows:

1.  **AIDA** contains $18,448$ training samples, $4791$ validation samples and $4485$ test samples. It also served as one of the largest manually annotated EL and ED datasets. Note that other datasets contain test split only.

2.  **MSNBC** is a news corpus with 20 documents and 656 mentions. Despite its small scale, MSNBC covers 10 obvious topics and thus acts as a perfect testbed for our topic VAE approach.

3.  **AQUAINT** is another news corpus containing 50 documents from Xinhua News, the New York Times and the Associated Press, covering 727 samples.

4.  **ACE2004** is a manually annotated subset of Doddington et al. (2004) containing 257 samples.

5.  **CWEB** is an automatically constructed dataset from ClueWeb corpus[4] in Guo and Barbosa (2018), containing 11,154 samples. It is worth mentioning that most ED works perform similarly on CWEB, partially because the average length of documents in CWEB is

---

[3] Specifically, prepositions refer to those that are frequently used in the category such as 'in', 'from', 'for', 'of', 'by', 'for', and 'involving'.

[4] https://lemurproject.org/clueweb12

significantly longer than others. With approximately 1,700 words on average per document, none of the trivial BERT-based models can handle the entire document. Yang et al. (2018) got relatively better performance on it as their models involve heavy hand-crafted features designed to capture document-level semantics. We later show that our topic learning-based methods can achieve similar performance with minimal human involvement.

6. **WIKI** is another automatically extracted test dataset from Gabrilovich et al. (2013), covering 6,821 samples.

## C Preparing Input Tokens for COHERENTED

First, the document is tokenized into $L_D$ text tokens and split into sentences. If text tokens fit in the word input window (i.e. $L_D \leq L - k - n_e$), we utilize all word tokens and sample $k$ topic sentences uniformly. Otherwise, we truncate the document with the sentence to be disambiguated as the center[5], and prioritize sampling $k$ sentences outside the trimming range as the topic sentences. Selected sentences get encoded through a pre-trained topic encoder and prepend their topic representations on the input sequence. Note that the topic decoder is no longer needed as it only supports variational learning in the auxiliary branch. Lastly, $N$ [MASK] tokens are appended to the sequence, indicating all $N$ mentions are unresolved. For training samples where $N < n_e$, we concatenate more $N - n_e$ [PAD] tokens.

## D Brief Introductions of Peers

- **AIDA** (Yosef et al., 2011) is a traditional framework and online tool for entity detection and disambiguation.

- **REL** (van Hulst et al., 2020) is a modern open-source toolkit for entity linking equipped with customized deep models.

- **GENRE** (Cao et al., 2021) is the first to formulate entity linking and disambiguation into the constrained text generation task via predefined trie. Auto-regressive decoding nature makes it hard for real-time usage.

---

[5]Different sampling strategies bring negligible improvement. As such, we will not discuss performance differences caused by sampling.

- **Bootleg** (Orr et al., 2021) focuses on modeling reasoning patterns for disambiguation in a self-supervised manner. Tail entities who rarely appear in KB and documents are especially investigated in this work.

- **BiBSG** (Yang et al., 2018) is the first to introduce the structured gradient tree boosting (SGTB) algorithm to collective entity disambiguation with many efforts in making use of global information from both the past and future to perform a better local search.

- **ExtEND** (Barba et al., 2022) formulates the ED problem into a span extraction task supported by a Longformer model that predicts entity span in the input sequence.

- **ReFinED** (Ayoola et al., 2022) is an efficient zero-shot end-to-end entity linker using score-based bi-encoder architecture, which seeks a trade-off between performance and efficiency.

- **GlobalED** (Yamada et al., 2022) considers ED as a masked token prediction problem and is also the baseline of our work.

## E Implementation, Training and Hyperparameters

| Hyperparameter | Value |
|---|---|
| learning rate (stage 1) | 5e-4 |
| learning rate (stage 2) | 5e-5 |
| weight_decay | 1e-2 |
| batch size per device | 4 |
| effective batch size | 2048 |
| learning rate strategy | WarmupDecayLR |
| optimizer | AdamW |
| dropout | 0.1 |
| gradient clipping | 1.0 |

Table 2: Hyperparameters used for training COHERENTED

We use the Huggingface (Wolf et al., 2020) version of Transformer as the codebase. A total of $127,314$ entities are considered in entity vocabulary and entity prediction head, resulting in a category vocabulary of size $391,234$. DeepSpeed (Rasley et al., 2020) is used for maximum hardware utilization and parallel training management. Table 2 presents most of the hyperparameters in training. Due to the limited computation

| Topic: Economics | |
| --- | --- |
| There were more declining shares than advancers on the New York Stock Exchange and the Nasdaq Stock Market ….Investors are looking for signs that consumer spending, one of the biggest drivers of the U.S. economy, will recover during the holidays. A report on industrial production weighed on the market. The Fed said output at the nation's factories, mines and utilities rose 0.1 percent in October, less than the 0.4 percent predicted by economists polled by Thomson Reuters… | |
| The Federal Reserve | Bank regulation in the United States, Central Banks… |
| The Fed (newspaper) | |
| The Federalist Papers | |
| The Federation (group) | Musical groups established in 2002, Hip hop groups from California… |

| Topic: Music |
| --- |
| …The song even induced a riot when The Fed performed "Hyphy" during halftime of the AND1 Live Tour at Oracle Arena in June 2004. On the strength of "Hyphy" and their second single "Donkey", the group's self-titled debut album was released under Virgin Records to critical reception… |

Figure 6: ED case analysis revealing two critical motivations of our work, topic coherence and categorical coherence. Mentions are annotated with a gray background. Two documents share the same mention "The Fed". The document in the upper part is centered around the economics topic while the lower one elaborates on the music topic. The middle part lists four entity candidates for the mention "The Fed", with some corresponding category labels on yellow background.

resource, we do not run massive hyperparameter searches. Multi-task coefficients are set to $\alpha = 0.1$ and $\gamma = 10$ with a few empirical trials done. During inference, the $k$ in Top-K category retrieval is set to 10, as the average number of categories of all entities present is close to this number.

We train our proposed model in two stages. In stage 1, we freeze all the parameters except for the fresh entity embedding layer and category memory layer. Consequently, the variational objective is disabled in stage 1. Then after 1 epoch, we activate all parameters and enable three objectives in stage 2, which lasts for 6 epochs. Pre-training a VAE can be difficult due to the notorious KL vanishing issue (Bowman et al., 2016), causing the decoder completely ignores the topic latent $z$ in learning. As a practical solution to mitigate this, a cyclical schedule is applied to the KL regularizer coefficient $\beta$. The training takes approximately 1 day for COHERENTED$_{base}$ and 3 days COHERENTED$_{large}$ on 8 A100-SXM4-40GB GPUs.

## F Case Study

In Figure 6, we illustrate document samples extracted from the MSNBC test dataset, wherein the mention "The Fed" can be readily disambiguated if provided with the corresponding topic.

Besides topic coherence, the relationship among entities also matters in the ED task. In the upper of Figure 6, mentions are highly correlated in their category level, and previous bi-encoder and cross-encoder solutions totally ignore the dependencies among entities and focus on learning representations alone.

Consider the upper document in Figure 6, where three mentions are highlighted with a gray background. The correct linked entity for the mention "New York Stock Exchange" shares the exact category "Stock exchanges in the United States" with the correct linked entity for "Nasdaq Stock Market." Moreover, these two mentions can implicitly guide entity prediction for "The Fed" due to the high correlation between their respective categories.