# OpenReview forum: "Coherent Entity Disambiguation via Modeling Topic and Categorical Dependency"
_EMNLP/2023/Conference — EMNLP 2023 Findings_

### Official Review · Reviewer_AHEW · 2023-07-19

**Typos Grammar Style And Presentation Improvements:** None
**Soundness:** 4

**Excitement:**

4: Strong: This paper deepens the understanding of some phenomenon or lowers the barriers to an existing research direction.

**Missing References:**

I believe the authors should also discuss similar usages of autoencoder architectures for topic/category embedding learning, such as the following two papers:
* Bianchi et al. “Pre-training is a Hot Topic: Contextualized Document Embeddings Improve Topic Coherence.” ACL 2020
* Meng et al. “Topic Discovery via Latent Space Clustering of Pretrained Language Model Representations.” WWW 2022

**Paper Topic And Main Contributions:**

This paper studies the entity disambiguation (ED) task and proposes a new framework CoherentED aiming to enhance high-level topical coherence in entity predictions. The method consists of the following modules:
* A variational autoencoder (VAE) that is used to extract latent topic vectors given input sentences. This ensures that topic coherence is preserved.
* Category memory embedding table that is inserted into the intermediate Transformer layers, enabling the system to retrieve relevant categories for undecided mentions when appropriate.

The authors conduct evaluations on several ED tasks and demonstrate that CoherentED generally outperforms baselines by around 1.3 F1 points.

**Questions For The Authors:**

None

**Reasons To Accept:**

* Neat and reasonable model design: The use of a VAE for topic embedding extraction and category memory augmentation make sense to me. They are conceptually simple and clear.
* Clear presentation: The paper is clear in its motivation and relevance to related work. The method part is introduced in detail clearly.
* Effectiveness: The proposed method is effective and outperforms compared baselines. There are also ablation studies that support the influence of each module along with visualizations of learned topic embeddings.

**Reasons To Reject:**

* Novelty: While the combination of existing methods forms a new framework CoherentED, the overall novelty of the paper appears to be not very strong. Specifically, the VAE architecture consisting of a BERT encoder and a GPT decoder is directly borrowed from Optimus (Li et al. 2020), and the masked entity-style training of the entity extractor is largely similar to LUKE (Yamada et al. 2020) and the ED model in Yamada et al. 2022.
* Lack of efficiency analysis: Since the CoherentED framework introduces an additional VAE for topic embedding extraction and category memory augmentation, it'll be necessary to analyze the latency brought by the extra modules (considering the baselines don't have these).

**Reproducibility:**

3: Could reproduce the results with some difficulty. The settings of parameters are underspecified or subjectively determined; the training/evaluation data are not widely available.

**Reviewer Confidence:**

3: Pretty sure, but there's a chance I missed something. Although I have a good feel for this area in general, I did not carefully check the paper's details, e.g., the math, experimental design, or novelty.

---

> ### Author Rebuttal · Authors · 2023-08-29
>
> We highly appreciate the reviewer's time and effort in providing feedback on our manuscript. And we would like to respond to the concerns raised, especially on the efficiency issue.
>
> ### Novelty Issue
>
> We would not deny that some individual components of CoherentED have precedents in previous works, such as the VAE architecture from Optimus and masked entity modeling from LUKE. These render our work less creative.
>
> However, the novelty of our framework lies in their unique combination of design motivation, with each of designs focusing on specific problems. For instance, the VAE design provides soft topic representations for later main model to condition on, providing global topic coherence. We also integrate a novel external category memory bank for intermediate entity representations to attend on, along with an oracle category label guidance strategy, which proves to be effective from evaluation results. `CoherentED` synthesizes these elements into a singular, optimized framework that exhibits synergy and improved performance over its constituent parts.
>
> ### Runtime Efficiency
>
> We would like to present the runtime efficiency comparison table on K50 consisting of 50 documents and 150 unresolved mentions.
>
> | Method | Basemodel | K50 Evaluation Took |
> | --- | --- | --- |
> | De Cao et al. (2021) | BART-large | 196.3 s |
> | Yamada et al. (2022) | BERT-large | 11.6 s |
> | Barba et al. (2022) | longformer-large-4096 | 5.72 s |
> | Our CoherentED$_{\text{large}}$ | Customized BERT-large | 15.9 s |
>
> Compared to the baseline solutions, `CoherentED` does suffer in terms of inference speed, but the overall efficiency is acceptable. Importantly, using the `CoherentED` solution not only yields more accurate entity disambiguation results, but the soft topic tokens produced by the VAE Encoder module and the category prediction obtained by querying the Category Memory can also serve as valuable by-products for downstream applications, such as for interpretability analysis.
>
> ### Missing References
>
> We greatly appreciate your mention of related works concerning autoencoder-based topic models, especially the numerical evaluation of topic clustering in Bianchi et al., which is highly relevant to the further investigation of the source of performance improvement suggested by Reviewer LkD8. Works related to topic modeling will be added to Section 2 *Related Works* in our next manuscript.
>
> Meng et al. focus more on unsupervised topic discovery using pre-trained language models.  While they did not seek to variational training objective, the strong and intuitive regularization loss terms helped the model to learn compact topic representations that can be verified by qualitative results and t-SNE visualization (which we also used in our work). We would add this work to Section 2 *Related Works* and build a qualitative topic probe to assess the topic modeling ability of our VAE architecture.
>
> Hope these would address all your concerns of our work! And thanks again for the constructive comments.

---

### Official Review · Reviewer_Wj22 · 2023-07-28

**Soundness:** 3

**Excitement:**

3: Ambivalent: It has merits (e.g., it reports state-of-the-art results, the idea is nice), but there are key weaknesses (e.g., it describes incremental work), and it can significantly benefit from another round of revision. However, I won't object to accepting it if my co-reviewers champion it.

**Paper Topic And Main Contributions:**

The paper is an extension of Yamada (2022) work with two new component: topic embeddings from neighbour sentences and entity category embeddings. For topic modelling, the paper uses a VAE whose encoder is BERT and decoder is GPT-2. The paper shows that with the two new components, new SoTA is achieved on average across 6 datasets (AIDA, MSNBC,...) without finetuning on AIDA training set.

**Questions For The Authors:**

q1: from eq1, it seems that z is a sequence of hidden vector z1...zn, and thus each sentence is encoded in a sequence of hidden zs rather than only one vector z. But it contradicts with line 250. What is true here?

q2: Eqs in sec 3.4 show that E2 is computed by a LayerNorm and a Transformer. How could it be?

q3: Line 404, it is quite unclear what "oracle category guidance" is. Could the authors explain it in a clearer way?

q5: Line 848, the model sample k sentences and "different sampling strategies bring negligible improvement". Why is sampling still used?

**Reasons To Accept:**

The paper points out two problems of current systems and fixes them with the two proposed components. The motivation is thus clear, and the solutions are backed by SoTA results on average across 6 popular datasets.

**Reasons To Reject:**

First of all, the paper is quite difficult to understand. Please see the *questions* section for what should be clarified.

Second, there can be some problems in the experiments.
* AIDA dataset is a standard for EL/ED evaluation in the literature. However, the paper does not present experiments for finetuning the proposed model on AIDA.
* The paper doesn't fully show Yamada (2022) results -- lacking performance on AIDA. Note that in Yamada (2022) show on table 1 the performance of their model w/o finetuning.
* The proposed model is trained on wikipedia only. The paper doesn't mention about any dev set, and thus it is unclear how hyper-params are tuned.

**Reproducibility:**

3: Could reproduce the results with some difficulty. The settings of parameters are underspecified or subjectively determined; the training/evaluation data are not widely available.

**Reviewer Confidence:**

3: Pretty sure, but there's a chance I missed something. Although I have a good feel for this area in general, I did not carefully check the paper's details, e.g., the math, experimental design, or novelty.

---

> ### Author Rebuttal · Authors · 2023-08-29
>
> We sincerely thank the reviewer for their insightful comments and valuable suggestions. We address each of the concerns below, with particular emphasis on fixing the readability issue.
>
> ### Readability Issue
>
> q1: from eq1, it seems that z is a sequence of hidden vector z1...zn, and thus each sentence is encoded in a sequence of hidden zs rather than only one vector z. But it contradicts with line 250. What is true here?
>
> a1: The true case is that $k$ topic sentences are encoded into topic representations that fit onto the start of the main model’s input as soft prompt tokens. We apologize for the mistake made in Section 3.3.1, where `input token sequence` confuses readers a lot. A more appropriate term would be `tokens of a topic sentence`.
>
> ---
>
> q2: Eqs in sec 3.4 show that E2 is computed by a LayerNorm and a Transformer. How could it be?
>
> a2:
>
> We apologize for the indexing error in Equation 3.4. The corrected third and fourth lines of Equation Group 3.4 are as follows, where $ \mathbf{E}^{1 \prime} $ represents the intermediate entity hidden states after aggregating category memory:
>
> $ \mathbf{E}^{1 \prime}=\operatorname{LayerNorm}\left(\mathbf{H}+\mathbf{E}^1\right) $
>
> $ \mathbf{T}^2, \mathbf{W}^2, \mathbf{E}^2=\text{Transformer}_{N}\left(\mathbf{T}^1, \mathbf{W}^1, \mathbf{E}^{1 \prime}\right)$
>
> ---
>
> q3: Line 404, it is quite unclear what "oracle category guidance" is. Could the authors explain it in a clearer way?
>
> a3: We would be happy to provide a more intuitive explanation.
>
> During the inference process, as the entity masks are predicted one at a time, there are no resolved entities at the beginning of inference. As a result, the intermediate representations corresponding to all entities must engage in attention operations with the external category memory layer to obtain the retrieved category representations as described in the second line of Section 3.4.
>
> However, once an entity has been finalized in the prediction, we will directly use the true category vector of this known entity as the attention coefficient, which aligns more closely with the supervision during the training process.
>
> ---
>
> q5: Line 848, the model sample k sentences and "different sampling strategies bring negligible improvement". Why is sampling still used?
>
> a5: When handling too long documents with `CoherentED`, one has to select $k$ topic sentences out of all $N$ sentences. Despite the performance consistency among different sampling strategies, sampling topic sentences **out of** the model’s context window is more intuitive with our motivation, that is to capture global topic coherence.
>
> ### Missing Yamada (2022) Results
>
> We double-checked the Yamada (2022) work and found that they reported **In-KB accuracy** on AIDA-test-B rather than micro F1 score we generally use. We will append reproduced Yamada (2022)’s AIDA F1 score w/o fine-tuning on AIDA onto Table 1 in future manuscript.
>
> | Method | AIDA | MSNBC | AQUAINT | ACE2004 | CWEB | WIKI | AVG-5 | AVG-6 |
> | --- | --- | --- | --- | --- | --- | --- | --- | --- |
> | Yamada et al. (2022) | - | 96.3 | 93.5 | 91.9 | 78.9 | 89.1 | - | 89.9 |
> | Reproduced Yamada et al. (2022) | 88.7 | 96.1 | 94.6 | 92.3 | 76.9 | 88.7 | 89.7 | 89.6 |
> | Our CoherentED$_{\text{large}}$ | 89.4 | 96.3 | 94.6 | 93.4 | 81.1 | 90.6 | 90.9 | 91.2 |
>
> ### Hyperparameter Tuning and Dev Set
>
> We apologize for the oversight of not introducing the strategy for hyper-parameter tuning. As we mentioned in Line 903-905, we do not run massive hyper parameter searches. We randomly held out 5% articles of Wikipedia 2022-09-01 dump and chose best-performing hyperparameter setting. The final checkpoint was trained with the full volume of the dump and the chosen hyperparameter setting.
>
> Thank you for your valuable feedback and we hope these could address most of your concerns!

---

### Official Review · Reviewer_LkD8 · 2023-08-06

**Soundness:** 3

**Excitement:**

3: Ambivalent: It has merits (e.g., it reports state-of-the-art results, the idea is nice), but there are key weaknesses (e.g., it describes incremental work), and it can significantly benefit from another round of revision. However, I won't object to accepting it if my co-reviewers champion it.

**Paper Topic And Main Contributions:**

This paper proposes an event disambiguation method equipped with two designs to enhance entity predictions' coherence. The authors use an unsupervised variational autoencoder (VAE) to extract latent topic vectors of context sentences, which are then used as extra inputs. Meanwhile, they incorporate an external category memory, enabling the method to retrieve relevant categories for undecided mentions, hence improving coherence at the category level. The method achieves competitive performance on popular ED benchmarks.


**Questions For The Authors:**

1. Does the semantics of topic and category partly overlap?
2. How does the number of topic tokens impact the overall performance?
2. In the VAE, what is the intuition of using BERT as the encoder and GPT2 as the decoder? Can an encoder-decoder pre-trained model work?


**Reasons To Accept:**

1. The two technical design is reasonable and presented clearly.
2. The experiments show the method's effectiveness, and the ablation study verifies that each key component contributes to the performance improvement.


**Reasons To Reject:**

1. The motivation is somewhat unclear to me. Generally, the authors claim that previous efforts can not handle entity interaction regarding topic coherence and category coherence. The authors can provide more intuitive examples by comparing existing methods, e.g., generative entity linking techniques.

2. The performance improvements are evident, but whether they are from better topic and category coherence is not verified. Figures 4 and 5's visualization just prove that the same or similar topics/categories will generate grouped embeddings, and more in-depth analyses are needed from the perspective of coherence.

3. The authors claim their model demonstrates outstanding performance in challenging long-text scenarios. More experiments and discussion on the context of different lengths can make this claim more convincing.


**Reproducibility:**

4: Could mostly reproduce the results, but there may be some variation because of sample variance or minor variations in their interpretation of the protocol or method.

**Reviewer Confidence:**

3: Pretty sure, but there's a chance I missed something. Although I have a good feel for this area in general, I did not carefully check the paper's details, e.g., the math, experimental design, or novelty.

---

> ### Author Rebuttal · Authors · 2023-08-29
>
> Firstly, we appreciate the time and effort you invested in reviewing our work. We have thoroughly considered each comment and would like to address the main concerns raised.
>
> ### Unclear Motivation
>
> We would like to emphasize that the Case Study is presented in Lines 926-946 of Appendix F. This study provides an example from our test dataset to clarify the motivation behind our model's design, which is to focus on topic and category coherence. We appreciate that Reviewer AHEW also values this. However, as the case study only includes predictions of our work, more intuitive side-by-side comparisons will be added to the future manuscript to show differences.
>
> It's worth noting that the method of Generative Entity Linking inherently incorporates dependencies on generated entities during the autoregressive process by attending to their text tokens of entity identifiers. This is fundamentally distinct from our CoherentED design, where **entities and categories are treated as independent and discrete elements** with no entity titles involved.
>
> ### The Source of Performance Improvement
>
> We acknowledge your concerns regarding the source of performance improvement. Indeed, Figures 4 and 5 can only demonstrate the representation learning capabilities of the model for topics and categories at the level of **embedding similarity**. We would like to point out that it is less likely for the number of model parameters to be the source of performance improvement, as we have discussed in Line 431-436.
>
> For the future manuscript, we will make the following adjustments:
>
> 1. A comparative Case Study and Error Analysis between `CoherentED` and previous works will be conducted, with special attention given to scenarios where `CoherentED `captures dependencies at the topic and category levels.
> 2. We plan to employ automated tools to design a category-level coherence evaluation metric for popular ED benchmarks, and then compare the baselines and `CoherentED`'s performance on category coherence. Specifically, leveraging the category system established in Appendix A, we can identify sets of categories that belong to known entities in the benchmark. The extent to which these sets overlap provides a measure of a document's sensitivity to category coherence. In this manner, `CoherentED`'s capability to capture category coherence can be quantified by comparing the F1 scores across separated test sets that group by category sensitivity.
> 3. We will utilize commonly used evaluation methods in topic modeling, such as Latent Dirichlet Allocation (LDA), to assess topic coherence. By comparing the UMass Coherence Score between sentence encoders trained with and without VAE, we can explore the strong correlations between real topic labels and topic tokens, on which the later model will condition.
>
> ### Case Study of Long-text Scenarios
>
> We understand your concerns regarding the lack of persuasive evidence for CoherentED's performance in long-text scenarios. In addition to our exceptional performance on CWEB, where the average document length exceeds 1700 tokens, we plan to report performance metrics for other benchmark subsets grouped by token length in future manuscripts. Corresponding case studies will also be added to Appendix F for further scrutiny.
>
> ### Responses for Questions
>
> Question 1: Does the semantics of topic and category partly overlap?
>
> Answer 1:
>
> Intuitively, topics and categories are likely to have overlapping semantics, as documents discussing similar topics are prone to containing entities with similar category labels. Due to the orthogonal designs in CoherentED, it becomes challenging to experimentally explore this aspect beyond what has been discussed in the section on 'The Source of Performance Improvement'. Investigating the interpretability of CoherentED will be an intriguing avenue for future work.
>
> ---
>
> Question 2: 1. How does the number of topic tokens impact the overall performance?
>
> Answer 2:
>
> The number of topic tokens we are using as the model’s prefix input is $k=8$. As the training took longer than the rebuttal period with our current compute resources, we are not able to give ablations only on $k$. Instead, we report our evaluation results from early experiment records in the table below.
>
> Note that hyperparameters $k$, $\alpha$ (loss coefficient for variational objective), and $\gamma$ (loss coefficient for category objective) are chosen using a held-out validation set from Wikipedia dump. Also early records **might not be reproducible** as the model went through changes over time.
>
> | k | $ \alpha $ | $ \gamma $ | AVG-6 |
> | --- | --- | --- | --- |
> | 8 | 0.1 | 10 | 90.9 |
> | 4 | 1e-2 | 1 | 89.9 |
> | 16 | 0.1 | 10 | 90.2 |
>
> ---
>
> Question 3: 1. In the VAE, what is the intuition of using BERT as the encoder and GPT2 as the decoder? Can an encoder-decoder pre-trained model work?
>
> Answer 3:
>
> The motivation of docking BERT and GPT-2 is to create an information bottleneck and expect the intermediate latent would carry light and compact representations for the main model to condition on. As Reviewer AHEW points out, this intuition originally comes from Optimus [1] which runs large-scale variational pre-training on corpus. Our work focuses more on producing sentence-level (k uniformly-sampled sentences) topic representations that could support specific usage.
>
> An encoder-decoder pre-trained model might work in a variational way with some dimension adaption and time-step aggregation layers between the encoder and decoder.
>
> ---
> Thank you again for your valuable feedback and we hope that our responses clarify any ambiguities.

---

### Meta-Review · Area_Chair_ZVKx · 2023-09-19

**Recommendation:** 3

**Metareview:**

This paper introduces CoherentED, an entity disambiguation system that incorporates topic embedding of context sentences extracted by VAE and an external category memory to retrieve relevant categories for undecided mentions. The approach design is reasonable and effective though the key components of CoherentED are borrowed from existing studies. Experimental results demonstrate promising improvements on 6 benchmark datasets. One main concern from most reviewers is the experiment analysis: more discussions about the performance on long documents, supervised performance on AIDA, latency, etc. are needed.

---

### Decision · Program_Chairs · 2023-10-07

**Decision:**

Accept-Findings

**Comment:**

This paper introduces CoherentED, an entity disambiguation system that incorporates topic embedding of context sentences extracted by VAE and an external category memory to retrieve relevant categories for undecided mentions. The approach design is reasonable and effective though the key components of CoherentED are borrowed from existing studies. Experimental results demonstrate promising improvements on 6 benchmark datasets. One main concern from most reviewers is the experiment analysis: more discussions about the performance on long documents, supervised performance on AIDA, latency, etc. are needed.